# Identifying predictors and determining mortality rates of septic cardiomyopathy and sepsis-related cardiogenic shock: A retrospective, observational study

Kathryn W. Hendrickson[1], Meghan M. Cirulis[2,3], Rebecca E. Burk[4], Michael J. Lanspa[2,3], Ithan D. Peltan[2,3], Hunter Marshall[3], Danielle Groat[2,5], Al Jephson[2,5], Sarah J. Beesley[2,3], Samuel M. Brown[2,3] *

1 The Oregon Clinic Department of Pulmonary, Critical Care, and Sleep Medicine East, Portland, OR, United States of America, 2 Division of Pulmonary and Critical Care Medicine, Intermountain Medical Center, Salt Lake City, UT, United States of America, 3 Division of Pulmonary and Critical Care Medicine, University of Utah, Salt Lake City, UT, United States of America, 4 Renown Medical Group Department of Pulmonary and Critical Care Medicine, Reno, NV, United States of America, 5 Intermountain Healthcare, Information and Analytics, Salt Lake City, UT, United States of America

☯ These authors contributed equally to this work.

* Samuel.Brown@imail.org

**Data Availability Statement:** No - some restrictions will apply; The dataset relevant to this study cannot be shared publicly as it contains

## Abstract

### Introduction

Septic shock is a severe form of sepsis that has a high mortality rate, and a substantial proportion of these patients will develop cardiac dysfunction, often termed septic cardiomyopathy (SCM). Some SCM patients may develop frank cardiac failure, termed sepsis-related cardiogenic shock (SeRCS). Little is known of SeRCS. This study describes baseline characteristics of patients with SCM and SeRCS compared to patients with septic shock without cardiac dysfunction. We compare clinical outcomes among SCM, SeRCS, and septic shock, and identify risk factors for the development of SCM and SeRCS.

### Methods

Septic patients admitted to the ICU with an echocardiogram obtained within 72 hours were included. Left ventricular ejection fraction of ≤55% was used to define SCM, and cardiac index ≤2.1 L/min/m2 among patients with SCM defined SeRCS. Machine learning was used to identify risk factors for development of SCM and SeRCS. Logistic regression was used to compare mortality among groups.

### Results

Among 1229 patients, 977 patients had septic shock without cardiac dysfunction, 207 had SCM, and 45 had SeRCS. In patients with septic shock, the strongest predictor for developing SCM and SeRCs was a prior history of cardiac dysfunction. Mortality did not significantly differ among the three groups.

potentially identifying patient data. Data access requests can be sent to the Intermountain Health IRB at IRB@imail.org or 801-408-1991.

**Funding:** This research was supported by the Intermountain Research and Medical Society Foundation (IRMF) in the form of a grant to SMB [837].

**Competing interests:** The authors have declared that no competing interests exist.

## Conclusions

SCM and SeRCS affect a minority of patients with septic shock, disproportionately affecting individuals with a history of cardiac disease. We did not identify a mortality difference associated with SCM or SeRCS. Additional work is needed to define further subtypes and treatment options for this patient population.

## Introduction

Sepsis is a systemic response to infection that causes organ failure and death by means of a dysregulated host response. It ranges from mild to severe, with septic shock representing the severe end of the spectrum. In the United States, sepsis is estimated to be a contributing factor or primary cause of death in up to 381,000 people annually [1]. Mortality among patients with septic shock is between 19 to 30% [2–4].

Cardiac dysfunction in sepsis and septic shock, termed septic cardiomyopathy (SCM), is common and develops in 40–70% of adult patients depending on the definition used [5, 6]. Most prior studies examining SCM were limited by small sample size. There is also no consensus definition, but SCM is often defined as a depressed left ventricular ejection fraction (LVEF) in the context of sepsis or septic shock [7]. Depending on the definition used, SCM is a morbid condition, with mortality ranging from 14–65% [7].

Some patients with SCM have frank cardiac failure, termed here sepsis-related cardiogenic shock (SeRCS), where the cardiac index (CI) is ≤2.1 L/min/m2 despite adequate volume resuscitation. The progression from SCM to SeRCS may be a physiologic transition from a purely distributive process to a mixed shock state, with ongoing vasoplegia complicated by low cardiac output. To date, data on SeRCS are limited to case reports and case series [8–10]. Without a thorough understanding of the natural history of SeRCS, optimal treatment pathways are unclear.

We therefore sought to describe the baseline characteristics and outcomes of patients with SCM and SeRCs within a septic shock cohort, and to identify risk factors for SCM and SeRCS among patients with septic shock.

## Methods

Starting in December 2017, we conducted a retrospective observational study of adult patients who had been treated for septic shock at Intermountain Medical Center (IMC) between November 5, 2007 and July 14, 2017. Data was collected between November 30, 2007 and February 28, 2023. Data was accessed between November 7, 2017 and February 28, 2023. IMC is a tertiary care center within an integrated healthcare system located in the Intermountain West. Although all data was de-identified, authors had access to information that could identify individual participants during or after data collection in order to complete the data analysis. IRB approval was obtained from the Intermountain Institutional Review Board; no study procedures or data queries were performed until after IRB approval had been received.

### Cohort definitions

First, eligible patients with sepsis who received vasopressors (i.e., those with septic shock) were identified via queries of the electronic data warehouse (EDW). Vasopressors were defined as norepinephrine, epinephrine, vasopressin, phenylephrine, or dopamine given 24 hours before

or after admission to the ICU and sustained for ≥24 hours. We used the Rhee adaptation of SEPSIS-3 to define sepsis: receipt of antibiotics in the presence of organ dysfunction [11, 12]. Among patients who met septic shock criteria, we included those who had an echocardiogram performed within 72 hours of ICU admission, a common practice at the study hospital. Exclusion criteria included evidence of obstructive or hemorrhagic shock, presence of ST segment myocardial infarction (STEMI), or admission for organ donation. Presence of hemorrhagic and obstructive shock were determined by manual chart review of patient records.

Among patients with septic shock, we identified patients with SCM defined as an LVEF ≤55%. If a patient previously had a low LVEF (<55%), SCM was defined as a decrease in LVEF of ≥10% (absolute) from baseline. Among patients meeting criteria for SCM, SeRCS was defined as patients with SCM who had at least one CI ≤2.1 L/min/m$^2$ as measured by echocardiogram.

## Data extraction

Data related to the septic shock, SCM, and SeRCS cohorts were manually gathered from medical charts and entered into REDCap [13, 14]. Demographic, clinical, laboratory, and treatment data for this study were obtained from the Intermountain Health EDW via queries [15]. Race and ethnicity were self-reported: non-Hispanic/Latino Black, Hispanic/Latino, non-Hispanic/Latino white, and Other. All labs and therapies were measured within the first 24 hours the patient met septic shock criteria. An indicator for high and/or low values was used for hemoglobin, platelet count, white blood cell count, and troponin. The LVEF (Simpson's method of discs) and CI were calculated by a critical care echocardiogram board-certified physician reader. The CI was calculated using the left ventricular outflow tract velocity time interval (LVOT-VTI) method [16]. The site of infection was defined by discharge ICD-10 codes.

All-cause mortality was defined at hospital discharge, 30 -days, and 90 -days, using the EDW, Utah State death records, and the Social Security Death Index.

## Comparison of baseline characteristics

Descriptive statistics were used to describe the prevalence and baseline characteristics of SeRCS, SCM, and septic shock within the overall cohort. Data are reported as median, (interquartile range), or count (percent) unless specified otherwise.

## Mortality analysis

Multivariable logistic regression was used to compare all-cause in-hospital mortality among the three subgroups. Predefined covariates included age, sex, race/ethnicity, highest SOFA score within the first 24 hours of ICU admission, Elixhauser Comorbidity Index using the Thompson 30 Index [17], highest norepinephrine equivalent (NEE) infusion rate within the first 24 hours of ICU admission, and receipt of mechanical ventilation at any time after ICU admission. Due to the exploratory nature of the analysis, we did not adjust for the multiple comparisons. Significance was set to p <0.05.

## Risk factor identification for SCM and SeRCS

We evaluated risk factors for both SCM and SeRCS among patients with septic shock. Features that were used for diagnostic purposes (e.g., LVEF and CI), and treatments and labs directly related to diagnosis (e.g. missingness of troponin was not at random) were not included in the modeling process. We took a 3-step approach to identify risk factos: 1) split data set into training and testing, 2) feature selection, and 3) model selection. Here we describe the data analysis

pipeline for risk factor identification of SCM within the septic shock cohort. A similar approach was used for SeRCS risk factors compared to the septic shock cohort, and differences are described below.

Prior to splitting the datasets, we used cut-off values to define abnormality of several lab results (S1 Table).

Next, the SCM and septic shock dataset was split at random for training (60%) and testing (40%) to create 100 versions of the data. In the feature selection step, the training data was prepared by normalizing continuous variables while categorical variables were encoded a binary variable for each unique category (i.e., one-hot coding). Three algorithms were applied to the training datasets, univariate logistic regression (LR), Random Forest (RF), and XGBoost (XGB) [18, 19]. For the RF analysis we turned three different hyperparameters: number of trees to grow, number of variables randomly sampled at each split, and minimum size of terminal nodes, in a grid-search fashion of 3 settings per hyperparameter for a total of nine models. Likewise, a total of nine XGBoost models were built with three settings for three hyperparameters: step size of each boosting step, maximum tree depth, and number of iterations.

P-values from univariate LR were used to order the features from lowest p-value to highest. RF features were ranked using the Gini score, or the mean decrease in accuracy if the feature is dropped from the tree. XGBoost used gain to set feature precedence, the average gain in accuracy across splits where the feature was used. Lists of features from the resulting models were created as ordered lists. In the case of predicting SCM from septic shock, the size of the feature list included up to 30 features while predicting SeRCS from septic shock included up to 10 features. Fewer features were used for SeRCS due to the limited number of observations for evaluation. The most commonly occurring feature set combination in the feature selection process was identified.

In the model selection step, each list of features and the test data were included in a multivariate logistic regression model. The best performing model with respect to the area under the curve (AUC) of the receiver operating characteristics curve (ROC) were presented as the prediction model for differentiating SCM or SeRCS. Additional metrics were reported on the best performing model, including accuracy, sensitivity, and specificity. Odds ratios were calculated for the best model and used to quantify the level of risk for the identified features. The best model was calibrated and provided a full report of its parameters for its potential use as a prediction model.

All analyses were conducted with R version 4.0.2 [20].

## Results

### Baseline characteristics

The initial query identified patients with septic shock and echocardiogram performed within 72 hours of admission during the study time frame. Exclusion criteria and data missingness required for diagnostic purposes excluded approximately 3,400 patients. We included 1229 patients with septic shock in this study, of which 977 (79%) had no SCM, 207 (17%) had SCM alone, and 45 (4%) had SCM with SeRCS (Fig 1). Compared to patients with septic shock alone, patients with SCM alone or SCM with SeRCS were similar age (63.0 [IQR 53.0–73.0]) vs 65.0 [IQR 52.0–76.5] vs 69.0 [IQR 53.0–76.0] but were less likely to be female (54% vs 42.0% vs 31.1%). SCM alone and SeRCS patients also had higher Elixhauser comorbidity scores (29.0 [IQR 18.0–41.0] vs 30.0 [IQR 19.0–44.0] vs 35.0 [IQR 25.0–47.0]) and more preexisting congestive heart failure (CHF) (35.4% vs 76.8% vs 93.3%). Mean LVEF was lower for SCM alone and SeRCS (65.0% [IQR 55.0–70.0%] vs 35.0% [IQR 26.5–41.5% vs 27.0% [IQR 18.0–

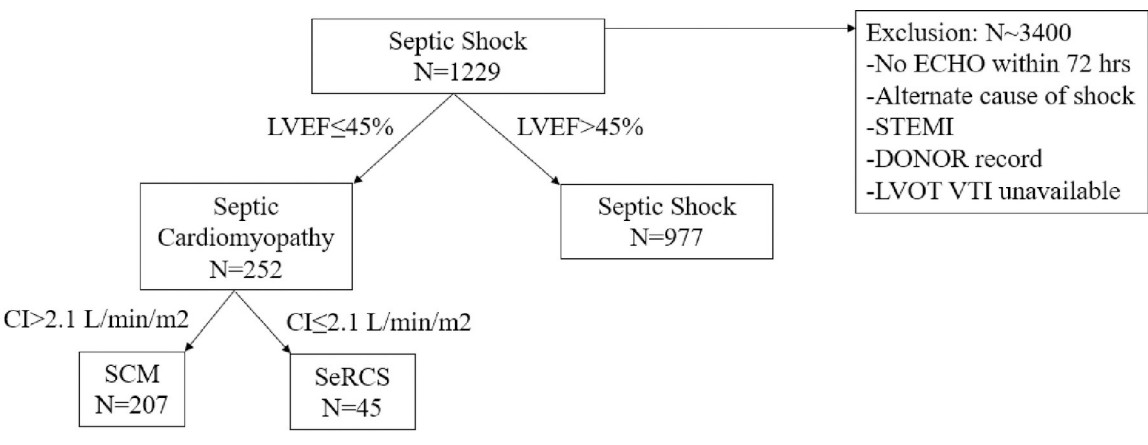

**Fig 1. Flow of patients into the study cohort.**

35.0%]). Inotropes (dobutamine or milrinone) were administered to 19.1% of patients with septic shock, 38.6% of patients with SCM alone, and 46.7% of patients with SeRCS (Table 1). Additional baseline laboratory results are presented in S2 and S3 Tables.

## Mortality

The all-cause in-hospital mortality rate for those with septic shock, SCM, and SeRCS was 386 (39.5%), 92 (44.4%), and 18 (40.0%) respectively. Mortality rates for 30-day, 90-day, and 1-year mortality followed a similar pattern within the SeRCS group (Table 2).

After controlling for relevant covariates, the presence of SCM (adjusted odds ratio [aOR] 1.27, 95% CI 0.91–1.76) or SeRCS (aOR 0.86, 95% CI 0.43–1.66) was not associated with increased in-hospital mortality compared to septic shock without SCM. Similarly, neither SCM or SeRCS were associated with increased risk-adjusted 30-days, 90-days, and 1-year mortality (Table 3).

## Identification of risk factors

The data analysis pipeline and machine learning algorithm (Fig 2) found he best performing features for predicting SCM were history of congestive heart failure (CHF), abnormal troponin, and male sex (Table 4, S4 Table). This model achieved an average AUC-ROC of 0.77. Logistic regression features and their coefficients are reported in Table 4.

The best performing model for SeRCs included history of CHF, history of valvular disease, history of coronary artery disease, and history of peripheral vascular disorders. The only other feature that occurred frequently but was not included in the final model was presence of a pacemaker (Table 4, S4 Table). The average AUC-ROC for the best performing model was 0.91. The accuracy was 0.85 with sensitivity and specificity maximized at 0.85 and 0.85, respectively. Feature coefficients from the logistic regression model are reported in Table 4.

In a post hoc analysis, we explored the natural history of SeRCS among the 3 patients with no prior history of cardiac dysfunction. Patient 1 was an intravenous (IV) drug user with tricuspid valve endocarditis, severe tricuspid regurgitation, and RV overload and failure. The patient had a history of coarctation of the aorta and bicuspid aortic valve with repair. One year prior to this admission he was admitted with tricuspid valve endocarditis with annuloplasty and completion of a full course of antibiotics. Echocardiogram before endocarditis confirmed LV valve function. He expired soon after admission to the ICU. Patient 2 had a history of

**Table 1. Patient demographics and baseline characteristics.** All data reported as median (interquartile range) or n(%) unless specified otherwise.

| Characteristic | Septic Shock without SCM | Septic Cardiomyopathy without SeRCS | SeRCS |
|---|---|---|---|
| | n = 977 | n = 207 | n = 45 |
| Age, years | 63.0 (53.0–73.0) | 65.0 (52.0–76.5) | 69.0 (53.0–76.0) |
| Female | 528 (54.0%) | 87 (42.0%) | 14 (31.1%) |
| Race/Ethnicity | | | |
| Non-Hispanic/Latino Black | 6 (0.6%) | 1 (0.5%) | 1 (2.2%) |
| Hispanic/Latino | 11 (1.1%) | 0 (0.0%) | 0 (0.0%) |
| Non-Hispanic/Latino white | 851 (87.1%) | 179 (86.5%) | 41 (91.1%) |
| Other | 109 (11.2%) | 27 (13.0%) | 3 (6.7%) |
| Body mass index (kg/m) | 29.0 (23.9–35.9) | 26.6 (22.9–31.4) | 28.9 (22.7–35.4) |
| Comorbity Scores | | | |
| Elixhauser Score | 29.0 (18.0–41.0) | 30.0 (19.0–44.0) | 35.0 (25.0–47.0) |
| APACHE II Score | 31.0 (24.0–39.0) | 31.0 (25.0–41.0) | 32.0 (25.0–41.0) |
| SOFA Score | 9.0 (7.0–11.0) | 9.0 (7.0–11.0) | 9.0 (7.0–12.0) |
| History of coronary artery disease | 386 (39.5%) | 119 (57.5%) | 37 (82.2%) |
| Pacemaker | 24 (2.5%) | 11 (5.3%) | 7 (15.6%) |
| Beta-blocker use | 148 (15.1%) | 31 (15.0%) | 24 (53.3%) |
| Blood Chemistries | | | |
| Creatinine | 1.8 (1.1–3.1) | 1.7 (1.1–2.6) | 1.8 (1.3–3.0) |
| Abnormal troponin | 623 (63.8%) | 167 (80.7%) | 39 (86.7%) |
| Abnormal BNP | 343 (35.1%) | 92 (44.4%) | 28 (62.2%) |
| High WBC | 536 (54.9%) | 97 (46.9%) | 30 (66.7%) |
| Low WBC | 196 (20.1%) | 55 (26.6%) | 7 (15.6%) |
| High hemoglobin | 42 (4.3%) | 6 (2.9%) | 3 (6.7%) |
| Low hemoglobin | 564 (57.7%) | 104 (50.2%) | 24 (53.3%) |
| High platelet | 75 (7.7%) | 8 (3.9%) | 2 (4.4%) |
| Low platelet | 354 (36.2%) | 67 (32.4%) | 16 (35.6%) |
| Lowest pH | 5.5 (5.0–6.0) | 5.5 (5.0–6.0) | 5.5 (5.0–7.0) |
| Highest norepinephrine equivalent rate (24 hours) | 0.2 (0.1–0.5) | 0.3 (0.1–0.6) | 0.3 (0.1–0.9) |
| Total norepinephrine equivalent administered (24 hours)(mcg/kg/min) | 2.3 (0.4–8.7) | 2.8 (0.5–8.4) | 2.4 (0.3–6.1) |
| Mechanically ventilated | 574 (58.8%) | 129 (62.3%) | 28 (62.2%) |
| Site of Infection | | | |
| Pulmonary | 75 (7.7%) | 17 (8.2%) | 8 (17.8%) |
| Urinary | 256 (26.2%) | 40 (19.3%) | 7 (15.6%) |
| Abdominal | 83 (8.5%) | 17 (8.2%) | 2 (4.4%) |
| Skin and soft sissue | 31 (3.2%) | 2 (1.0%) | 2 (4.4%) |
| Central nervous system | 8 (0.8%) | 0 (0.0%) | 0 (0.0%) |
| Bloodstream | 11 (1.1%) | 5 (2.4%) | 0 (0.0%) |

**Table 2. Mortality outcomes across subgroups.**

| Timepoint | Septic Shock | Septic Cardiomyopathy | SeRCS | Overall |
|---|---|---|---|---|
| | n = 977 | n = 207 | n = 45 | n = 1229 |
| All-cause in hospital | 386 (39.5%) | 92 (44.4%) | 18 (40.0%) | 496 (40.4%) |
| 30-day | 429 (43.9%) | 96 (46.4%) | 22 (48.9%) | 547 (44.5%) |
| 90-day | 484 (49.5%) | 107 (51.7%) | 24 (53.3%) | 615 (50.0%) |
| 1-year | 533 (54.6%) | 120 (58.0%) | 25 (55.6%) | 678 (55.2%) |

**Table 3. Adjusted odds ratio (95% confidence interval) for mortality at different time points for septic cardiomyopathy (SCM) and sepsis-related cardiogenic shock (SeRCS) compared to septic shock without SCM.**

| Timepoint | Odds Ratio (95% CI) | p-value |
|---|---|---|
| All-cause in-hospital mortality | | |
| SCM | 1.27 (0.91–1.76) | 0.160 |
| SeRCS | 0.86 (0.43–1.66) | 0.650 |
| 30-day Mortality | | |
| SCM | 1.12 (0.8–1.55) | 0.520 |
| SeRCS | 1.01 (0.52–1.96) | 0.980 |
| 90-day Mortality | | |
| SCM | 1.08 (0.78–1.5) | 0.650 |
| SeRCS | 0.92 (0.47–1.81) | 0.810 |
| 1-year Mortality | | |
| SCM | 1.15 (0.82–1.61) | 0.419 |
| SeRCS | 0.81 (0.41–1.59) | 0.527 |

coronary artery bypass grafting and stent placement in the coronary arteries, but no evidence in the HER to suggest heart failure or reduced LVEF. Patient 3 had end-stage liver disease and was admitted with empyema and septic shock. Echocardiogram three months prior to admission was unremarkable, while admission echocardiogram showed decreased LV and RV systolic function as well as new regional wall motion abnormalities. The patient expired soon after admission the ICU. Whether the regional wall motion abnormalities suggest obstructive coronary disease is unknown.

## Discussion

In this large, single-center cohort study we describe for the first time the epidemiology of SeRCS. SeRCs appears to be overwhelmingly the syndrome of septic shock in a subset of

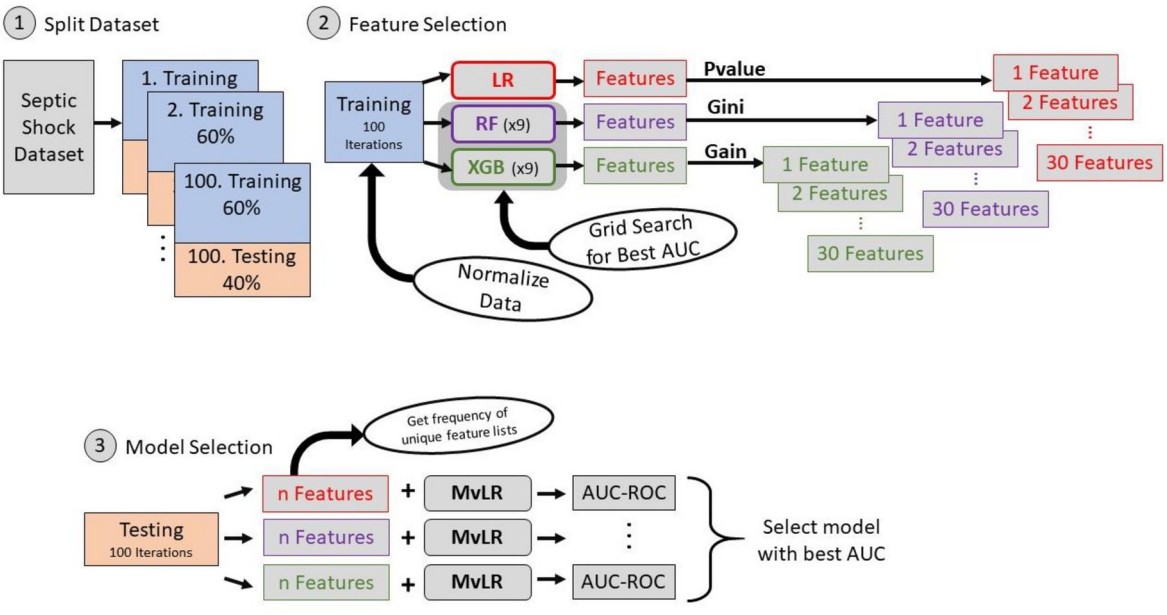

**Fig 2. Data analysis pipeline and machine learning algorithms to identify risk factors for development of SCM and SeRCS.**

**Table 4. Odds ratios and log odds coefficients from the best performing models to predict septic cardiomyopathy (SCM) and sepsis-related cardiogenic shock (SeRCS) within a cohort of patients with septic shock.**

| Feature | Odds Ratio | Log Odds | Threshold |
|---|---|---|---|
| SCM | | | 0.1293 |
| Intercept | - | -3.234 | |
| History of CHF | 4.07 | 1.40 | |
| Abnormal troponin | 2.30 | 0.83 | |
| Male | 1.63 | 0.49 | |
| SeRCS | | | 0.0779–0.1329 |
| Intercept | - | -5.367 | |
| History of CHF | 8.87 | 2.18 | |
| History of valvular disease | 2.02 | 0.70 | |
| History of coronary artery disease | 1.95 | 0.67 | |
| History of peripheral vascular disorders | 1.90 | 0.64 | |

patients with preexisting severe cardiac disease. These patients have similar outcomes, in the contemporary clinical environment (which commonly includes inotrope therapy), as patients with septic shock with or without SCM. A very small group of patients with SeRCS and no apparent preexisting cardiac dysfunction were identified, although in each case cardiac dysfunction independent of sepsis was a distinct possibility.

All risk factors for SeRCS identified in this cohort study related to prior cardiac history. Biomarkers associated with SeRCS include elevated troponin and elevated BNP, traditional markers of cardiac dysfunction.

Our study showed a lower rate of SCM compared to previous studies. This is likely due to two factors. First, our cohort is larger than prior cohorts and benefits from a common clinical practice in the study hospital of performing echocardiography in patients with shock, including septic shock. Second, we used a simple LVEF-based definition of SCM, whereas definition based in longitudinal strain may identify more patients with subtler cardiac dysfunction. Our results may thus better reflect a real-world population.

Similar to our study, previous studies of SCM based on LVEF measurements suggest that these patients did not have higher mortality than septic shock patients [8, 21]. Different definitions of SCM have been associated with different estimates of attributable mortality [22–25]. We chose the readily available and reproducible LVEF. We considered the possibility that inotrope therapy may have masked mortality differences, but septic shock patients with normal cardiac function received inotrope therapy at roughly similar rates to SCM and SeRCS patients (reflecting the then-common practice of goal-directed hemodynamic therapy in septic shock).

Similar to previous cohorts with SCM [7, 22], male patients with prior cardiovascular disease are most at risk for developing both SCM and SeRCS.

This, the largest study of SeRCS to date, lays the groundwork for future investigations into the optimal definition of sub-populations and treatment strategies for patients experiencing cardiogenic shock in the setting of septic shock. A limitation of this study was that it is single centered with a relatively small number of SeRCS patients. This definition may over-simplify SCM, as diastolic and right ventricular (RV) dysfunction have also been identified as components of the SCM syndrome [26, 27]. Given the limitations of the retrospective nature of this study Takotsubo phenotype was not specifically identified and it is possible the condition represents a portion of the SCM and SeRCS cohorts. Over the course of the study duration, much has changed in the diagnosis and treatment of sepsis and septic shock in critical care which increases the possibility of non-random biases influencing key parameters and outcomes of

this study. A key limitation of this and prior research among critically ill patients is the lack of perfect knowledge of the patient's comorbid state immediately prior to onset of sepsis. Some patients who appear to have septic cardiomyopathy may in fact have another etiology of their cardiac dysfunction. As an improvement upon other studies in this field, we did clinically review pre-admission echocardiograms. The security of our observed findings is thus higher than in many other studies in this field, but large, prospective, community-based cohorts would be required to answer this question definitively. SeRCS in patients without prior cardiac disease appears extremely rare in this cohort; defining epidemiology, natural history, and optimal treatment strategies in this group will require large-scale, multi-center collaborations.

## Conclusion

This study is the first to describe patients with SeRCS from a large cohort of ICU patients with septic shock. SeRCS patients largely have preexisting cardiac disease which deteriorates in the setting of septic shock. The patients with SeRCS have similar outcomes, even when treated with goal-directed hemodynamic therapy, as patients with SCM or septic shock alone. Other important risk factors for development of SCM with and without SeRCS are history of cardiovascular disease and male sex. Further study in this area is needed to clearly define SCM with and without SeRCS, and to delineate optimal treatments for this group of patients.

## Supporting information

**S1 Table. Cut-off values used to determine abnormal lab values.**
(DOCX)

**S2 Table. Additional baseline characteristics, reported as median (interquartile range) or count (percent).**
(DOCX)

**S3 Table. Gram stain results for blood, respiratory, and urine cultures of patients with positive culture results.**
(DOCX)

**S4 Table. Most frequently occurring features sets, where the feature set length was greater than one for predicting septic cardiomyopathy (SCM) and sepsis-related cardiogenic shock (SeRCS).**
(DOCX)

## Author Contributions

**Conceptualization:** Rebecca E. Burk, Michael J. Lanspa, Sarah J. Beesley, Samuel M. Brown.

**Data curation:** Meghan M. Cirulis, Rebecca E. Burk, Michael J. Lanspa, Hunter Marshall, Danielle Groat, Al Jephson, Sarah J. Beesley, Samuel M. Brown.

**Formal analysis:** Kathryn W. Hendrickson, Meghan M. Cirulis, Hunter Marshall, Danielle Groat, Al Jephson, Samuel M. Brown.

**Funding acquisition:** Rebecca E. Burk, Michael J. Lanspa, Sarah J. Beesley, Samuel M. Brown.

**Investigation:** Kathryn W. Hendrickson, Meghan M. Cirulis, Rebecca E. Burk, Michael J. Lanspa, Ithan D. Peltan, Hunter Marshall, Danielle Groat, Al Jephson, Sarah J. Beesley, Samuel M. Brown.

**Methodology:** Kathryn W. Hendrickson, Meghan M. Cirulis, Rebecca E. Burk, Michael J. Lanspa, Ithan D. Peltan, Danielle Groat, Al Jephson, Sarah J. Beesley, Samuel M. Brown.

**Supervision:** Samuel M. Brown.

**Validation:** Kathryn W. Hendrickson, Hunter Marshall, Sarah J. Beesley, Samuel M. Brown.

**Writing – original draft:** Kathryn W. Hendrickson, Danielle Groat.

**Writing – review & editing:** Kathryn W. Hendrickson, Meghan M. Cirulis, Rebecca E. Burk, Michael J. Lanspa, Ithan D. Peltan, Danielle Groat, Sarah J. Beesley, Samuel M. Brown.

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
