## [Decision Letter · Decision Letter 0]

7 Nov 2023

PONE-D-23-31117Identifying predictors and determining mortality rates of septic cardiomyopathy and sepsis-related cardiogenic shock: A retrospective, observational studyPLOS ONE

Dear Dr. Hendrickson,

Thank you for submitting your manuscript to PLOS ONE. After careful consideration, we feel that it has merit but does not fully meet PLOS ONE’s publication criteria as it currently stands. Therefore, we invite you to submit a revised version of the manuscript that addresses the points raised during the review process.

 Thanks Vikash Jaiswal, MD

We look forward to receiving your revised manuscript.

Kind regards,

Vikash Jaiswal, MD

Academic Editor

PLOS ONE

Journal Requirements:

Reviewers' comments:

Reviewer's Responses to Questions

**Comments to the Author**

1. Is the manuscript technically sound, and do the data support the conclusions?

Reviewer #1: Yes

Reviewer #2: Yes

2. Has the statistical analysis been performed appropriately and rigorously? 

Reviewer #1: Yes

Reviewer #2: Yes

3. Have the authors made all data underlying the findings in their manuscript fully available?

Reviewer #1: Yes

Reviewer #2: Yes

4. Is the manuscript presented in an intelligible fashion and written in standard English?

Reviewer #1: Yes

Reviewer #2: Yes

5. Review Comments to the Author

Reviewer #1: I would like to congratulate the authors for this remarkable work on septic cardiomyopathy and Sepsis induced cardiogenic shock. It is an important and yet not an well understood.

-Inclusion of data of over 10 years duration has added a lot of granularity in the data.

-The authors have investigated the association of clinical factors and outcome variables using univariate and multivariate logistic regression models.

-Is it possible to look into Takotsubo phenotype among the SCM group. It is quite understandable that it may not be logistically feasible given the retrospective nature of the study design.

-Since the 2000s there has been evolution in critical care practices. The rise and fall of the fluid resuscitation in sepsis and septic shock. There has been growing attention to minimization of sedation, early mobilization, and sepsis recognition and treatment, the latter of which may mitigate mortality due to sepsis. Because of these changes in practices it could have potentially impacted the clinical outcomes of interest. It would be important to acknowledge this in the manuscript specially given the retrospective nature of the study design.

Reviewer #2: The study's endeavor to elucidate the characteristics and implications of SCM is commendable. However, I have some concerns regarding the operational definition of SCM as presented in the manuscript, particularly concerning the use of echocardiographic findings to diagnose SCM.

The authors defined SCM as a condition in any patient presenting with sepsis who was found to have an ejection fraction (EF) < 55% or a reduction of > 10% from a previous echocardiogram. Although the intent to establish clear diagnostic criteria is appreciated, this definition poses potential issues that must be addressed to ensure the accuracy and reliability of the study’s conclusions.

The main issue arises from the assumption that a decrease in EF or an EF below 55% is attributable to sepsis. This was predicated on the availability of a recent echocardiogram for comparison. In the absence of a recent echocardiogram (i.e., one conducted shortly before sepsis onset), there is a substantial risk of misclassification. Specifically, if the patient had no echocardiographic evaluation before the current episode or the last echocardiogram was conducted more than six months ago, the decrease in EF or the presence of an EF < 55% might reflect pre-existing heart failure or other cardiac conditions unrelated to sepsis.

The heterogeneity of the patient's previous cardiac evaluations and the potential for a significant interval since the last echocardiogram might introduce a confounding variable that could skew the results and interpretation of the study.

I suggest the following revisions to the study design and analysis to mitigate these concerns.

1. Clearly define the time frame within which a previous echocardiogram must have been conducted to be considered valid for comparison. Considering the natural progression of cardiac function over time, a consensus on what constitutes an appropriate 'baseline' echocardiogram should be established.

2. In the absence of a recent echocardiogram, additional diagnostic criteria should be employed to differentiate SCM from preexisting cardiac conditions. This could include markers of cardiac strain related to sepsis, biomarkers, or a comprehensive review of the patient’s medical history of documented cardiac issues.

3. Conduct sensitivity analyses to determine how including patients without recent echocardiograms might affect the study's findings and conclusions. This could help to understand the potential bias introduced by including these patients.

Given the complexity of the septic milieu and its multifactorial impact on cardiovascular function, the attribution of LVD solely to sepsis, without rigorous longitudinal cardiac assessment, may lead to oversimplification of the pathophysiology. This is particularly pertinent in light of the authors' admission that a "very small group of patients with SeRCS and no apparent preexisting cardiac dysfunction were identified," which indicates that the occurrence of pure sepsis-induced cardiomyopathy is potentially much less common than previously thought.

6. PLOS authors have the option to publish the peer review history of their article (what does this mean?). If published, this will include your full peer review and any attached files.

Reviewer #1: No

Reviewer #2: No

---

## [Author Response · Author response to Decision Letter 0]

3 Jan 2024

Reviewer #1

1. Inclusion of data of over 10 years duration has added a lot of granularity in the data.

This is an important point. To enroll a sufficient number of patients to make meaningful conclusions required a long look-back period. Fortunately, IHC has a decades-long history of maintaining a robust electronic data warehouse which greatly reduces the granularity one would expect to find in other health systems.

2. Is it possible to look into Takotsubo phenotype among the SCM group. It is quite understandable that it may not be logistically feasible given the retrospective nature of the study design.

This comment has been taken to heart. Unfortunately, we would not be able to accurately review all records to make the diagnosis of Takotsubo phenotype given limitations in data available in this retrospective study. We have added to the last paragraph of the discussion section:

“Given the limitations of the retrospective nature of this study Takotsubo phenotype was not specifically identified and it is possible the condition represents a portion of the SCM and SeRCS cohorts.”

3. Since the 2000s there has been evolution in critical care practices. The rise and fall of the fluid resuscitation in sepsis and septic shock. There has been growing attention to minimization of sedation, early mobilization, and sepsis recognition and treatment, the latter of which may mitigate mortality due to sepsis. Because of these changes in practices it could have potentially impacted the clinical outcomes of interest. It would be important to acknowledge this in the manuscript specially given the retrospective nature of the study design.

To address this important point, we have added to the final paragraph of the discussion section: 

“Over the course of the study duration, much has changed in the diagnosis and treatment of sepsis and septic shock in critical care which increases the possibility of non-random biases influencing key parameters and outcomes of this study.”

Reviewer #2

1. Clearly define the time frame within which a previous echocardiogram must have been conducted to be considered valid for comparison. Considering the natural progression of cardiac function over time, a consensus on what constitutes an appropriate 'baseline' echocardiogram should be established.

The reviewer rightly notes an ongoing problem with all cohorts defined by ICU admission. The methods employed in our study are an improvement upon other work in the field, but the reviewer is correct that limitations remain. We have added the following text to the manuscript to acknowledge and engage this reviewer’s concern. We have included this limitation in the final paragraph of the discussion section:

“A key limitation of this and prior research among critically ill patients is the lack of perfect knowledge of the patient’s comorbid state immediately prior to onset of sepsis. Some patients who appear to have septic cardiomyopathy may in fact have another etiology of their cardiac dysfunction. As an improvement upon other studies in this field, we did clinically review pre-admission echocardiograms. The security of our observed findings is thus higher than in many other studies in this field, but large, prospective, community-based cohorts would be required to answer this question definitively.”

2. In the absence of a recent echocardiogram, additional diagnostic criteria should be employed to differentiate SCM from preexisting cardiac conditions. This could include markers of cardiac strain related to sepsis, biomarkers, or a comprehensive review of the patient’s medical history of documented cardiac issues.

This point is well taken. We analyzed cardiac and biomarkers as well as reviewed charts to exclude patients who had presented with a primary cardiac abnormality rather than septic shock. All patients in the study had septic shock as their primary pathology and given the limitations of a retrospective study, the changes in LVEF were assigned to the septic shock. Please see our comment above for the additional text added to the manuscript.

3. Conduct sensitivity analyses to determine how including patients without recent echocardiograms might affect the study's findings and conclusions. This could help to understand the potential bias introduced by including these patients.

As we have mentioned, obtaining baseline echocardiograms is a limitation of this study. However, the aim of this study was to create a prediction tool that would assist clinicians in identifying patients at risk for SCM and SeRCS. This objective necessitates the use retrospective data that would closely resemble the data a clinician would have access to during an encounter. In many cases, prior echocardiograms are not available to the clinician, nor are there existing tools that would help them to quickly compare a current echocardiogram to a prior one. Without having a reliable method to tease out these patients, we are not able to run a sensitivity analysis. Additionally, this patient cohort is not large and adding an additional subgroup to this cohort would likely not result in significant findings.

---

## [Decision Letter · Decision Letter 1]

19 Feb 2024

Identifying predictors and determining mortality rates of septic cardiomyopathy and sepsis-related cardiogenic shock: A retrospective, observational study

PONE-D-23-31117R1

Dear Dr. Hendrickson,

We’re pleased to inform you that your manuscript has been judged scientifically suitable for publication and will be formally accepted for publication once it meets all outstanding technical requirements.

Kind regards,

Vikash Jaiswal, MD

Academic Editor

PLOS ONE

Additional Editor Comments (optional):

Reviewer recommended acceptance.

Reviewers' comments:

Reviewer's Responses to Questions

**Comments to the Author**

1. If the authors have adequately addressed your comments raised in a previous round of review and you feel that this manuscript is now acceptable for publication, you may indicate that here to bypass the “Comments to the Author” section, enter your conflict of interest statement in the “Confidential to Editor” section, and submit your "Accept" recommendation.

Reviewer #1: All comments have been addressed

2. Is the manuscript technically sound, and do the data support the conclusions?

Reviewer #1: Yes

3. Has the statistical analysis been performed appropriately and rigorously? 

Reviewer #1: Yes

4. Have the authors made all data underlying the findings in their manuscript fully available?

Reviewer #1: Yes

5. Is the manuscript presented in an intelligible fashion and written in standard English?

Reviewer #1: No

6. Review Comments to the Author

Reviewer #1: The R1 version of the referenced manuscript is appropriate for publication. The previously suggeted review points have been amended by the authors.

7. PLOS authors have the option to publish the peer review history of their article (what does this mean?). If published, this will include your full peer review and any attached files.

Reviewer #1: **Yes: **None
